# Herd-Level and Individual Differences in Fecal Lactobacilli Dynamics of Growing Pigs

**DOI:** 10.3390/ani11010113

**Published:** 2021-01-07

**Authors:** Emilia König, Virpi Sali, Paulina Heponiemi, Seppo Salminen, Anna Valros, Sami Junnikkala, Mari Heinonen

**Affiliations:** 1Department of Production Animal Medicine, Faculty of Veterinary Medicine, University of Helsinki, 04920 Saarentaus, Finland; mari.heinonen@helsinki.fi; 2Functional Foods Forum, Faculty of Medicine, University of Turku, 20014 Turku, Finland; pebhep@utu.fi (P.H.); seppo.salminen@utu.fi (S.S.); 3Research Centre for Animal Welfare, Department of Production Animal Medicine, Faculty of Veterinary Medicine, University of Helsinki, 00790 Helsinki, Finland; anna.valros@helsinki.fi; 4Department of Veterinary Biosciences, Faculty of Veterinary Medicine, University of Helsinki, 00790 Helsinki Finland; sami.junnikkala@helsinki.fi

**Keywords:** lactobacilli, pigs, dynamics, count, longitudinal, intestinal microbiota, colostrum

## Abstract

**Simple Summary:**

Selection for hyper-prolific sows has led to increased litter size, decreased birth weight, and increased within-litter variation. This is accompanied by impaired colostrum intake of piglets and poor performance. We aimed to investigate the total count of fecal lactobacilli and species diversity in growing pigs on two herds. Study pigs were categorized either small or large according to their birth weight. Sow colostrum quality and colostrum supply of piglets were determined. We hypothesized that the birth weight and growth performance of pigs are associated with fecal lactobacilli composition, which is influenced by colostrum. Small pigs had higher lactobacilli counts in both herds, but the difference was significant only for one herd (*p* = 0.01). Colostrum quality was numerically better in the herd that appeared also better managed in comparison to the other study herd. Colostrum intake tended to be significantly associated with the total lactobacilli count in the better-managed herd. In conclusion, herd-level factors clearly contribute to the microbiota of pigs, but birth weight also plays a potential role in the gastrointestinal tract lactobacilli dynamics. Our results revealed a potential long-term effect of colostrum, and therefore give a reason to investigate more thoroughly the associations between maternal immunity, pig microbiota, and performance.

**Abstract:**

We studied the fecal lactobacilli count and species diversity of growing pigs along with immune parameters associated with intestinal lactobacilli. Thirty pigs categorized as small (S, *n* = 12) or large (L, *n* = 18) at birth were followed from birth to slaughter in two commercial herds, H1 and H2. Herds differed in terms of their general management. We determined sow colostrum quality, colostrum intake, piglet serum immunoglobulins, and pig growth. We took individual fecal samples from pigs in the weaning and finishing units. We studied lactobacilli count and identified their diversity with 16S PCR. Total lactobacilli count increased in H1 and decreased in H2 between samplings. Lactobacilli species diversity was higher in H1 in both fecal sampling points, whereas diversity decreased over time in both herds. We identified altogether seven lactobacilli species with a maximum of five (one to five) species in one herd. However, a relatively large proportion of lactobacilli remained unidentified with the used sequencing technique. Small pigs had higher lactobacilli counts in both herds but the difference was significant only in H2 (*p* = 0.01). Colostrum quality was numerically better in H1 than in H2, where colostrum intake tended to be associated with total lactobacilli count (*p* = 0.05).

## 1. Introduction

The first microbial colonizers of a piglet gastrointestinal tract (GIT) originate from the sow birth canal during parturition and from the pen environment [1,2]. The sow placenta is thought to be impermeable for microbes [3], contrary to results on cattle, horses, and humans, which all suggest in utero microbial gut colonization of the fetus [4,5,6]. Colostrum is one of the most powerful external factors affecting piglet microbiota development during the suckling period [3,7]. Several studies have positively associated colostrum intake (CI) with birth weight [8,9], and small piglets are consequently at higher risk of suffering from inadequate CI and subsequent impaired gut microbiota development.

The selection for hyper-prolific sow traits has led to an increase in litter size, and the average birth weight of piglets has consequently decreased [10], within-litter variation in birth weight has been reported [10], and neonatal mortality has increased [11] during recent decades. Additionally, low colostrum intake is one of the main reasons for poor piglet survival and poor growth after birth [12,13,14]. In addition to colostrum and milk providing essential energy for piglet growth, their contents are known to contribute to the GIT bacterial profile richness of pigs [7]. Bacterial phylogenetic diversity changes gradually from birth over the weaning period [7,15,16,17]. It is important to achieve an adult-like stable microbiota that is more resistant to pathogens as early as possible [3] while still being capable of successfully adapting and responding to environmental factors [18].

Antimicrobials are widely used in pig production, especially in young piglets [19]. The increasing problem of antimicrobial resistance (AMR) has motivated researchers to investigate potential alternatives for antimicrobials such as probiotics, prebiotics, and organic acids [20]. The International Scientific Association for Probiotics and Prebiotics (ISAPP) has defined probiotics as “live microorganisms that, when administered in adequate amounts, confer a health benefit to the host” [21]. Among the above-mentioned alternatives to antimicrobials, probiotics appear to be one promising option [22,23] for supporting the health and well-being of pigs. This is based on the capability of probiotics to resist microbial infections [24] and stabilize the gut microbiota [22,23]. The results of Dowarah et al. [25] suggest species-specific probiotics for reaching better results compared to probiotics retrieved from other animals.

Lactobacilli are a thoroughly researched genus of Gram-positive bacteria. Several *Lactobacillus* species are shown to have probiotic properties [23,26]. Further, many *Lactobacillus* probiotics have the notification “generally regarded as safe” (GRAS) issued by the United States Food and Drug Administration (FDA) [23,26] and also fall under “qualified presumption of safety” (QPS) in the European Union (EU) [27]. Lactobacilli are abundant in the GIT of pigs [26], thus urging investigations of their potential as pig-specific probiotics. To the knowledge of the authors, only a few studies have investigated the dynamics of fecal *Lactobacillus* species of piglets born as either small (S) or large (L) [17,28].

The objective of our study was to investigate the total lactobacilli count and lactobacilli species diversity in growing pigs categorized either as S or L according to their body weight at birth. To investigate factors associated with the fecal lactobacilli contents, we determined sow colostrum quality and measured piglet serum immunoglobulins as an indicator for colostrum supply, and calculated pig growth. We hypothesized that the birth weight and growth performance of pigs are associated with GIT lactobacilli composition, which is influenced by colostrum quality. Additionally, the results of this pilot study will provide information for identifying potential probiotic lactobacilli in pig feces for further screening.

## 2. Materials and Methods

### 2.1. Ethical Approval

The experiment was approved by the southern Finland Regional State Administrative Agency (ESAVI/16950/2018).

### 2.2. Herd and Animal Selection for Original Sampling

Two commercial herds, hereby abbreviated H1 and H2, located in western and south-western Finland, participated voluntarily in the study. Herds H1 and H2 averaged 450 and 1100 sows, respectively. Sows in both herds were housed in standard farrowing pens with crates and partly slatted floors. Both herds had separate finishing units to which the pigs were transported at the approximate age of ten weeks (Figure 1). The distance between the piglet producing unit and the finishing unit was 0.5 km for H1 and 320 km for H2, respectively. In general, the piglet-producing unit of H1 appeared better managed, with e.g., a lower percentage of piglet mixing and better hygiene, as assessed by the researchers. On the contrary, the finishing unit of H1 appeared to have lower hygiene than the H2 finishing unit. Both herds followed the routine Finnish vaccination program for sows and piglets. As a routine management procedure, piglets were castrated before seven days of age and iron supplementation was given concurrently. For analgesia, all piglets received one dose of a non-steroidal anti-inflammatory drug prior to castration and another dose the following day. No teeth clipping or tail docking was performed. Feeding was according to Finnish standards in all phases. Liquid feeding was used in both herds during all phases, except for the weaning unit in H2, where dry feeding was performed. Only dry feed was available ad libitum. No oral antimicrobial agents or growth promoters were given to any of the study pigs in the feed.

One farrowing group per herd was included in the study. Eleven sows (DanBred × Landrace) gave birth to 126 piglets within four days in H1 and 254 piglets were born from 18 sows (Topigs Norsvin) within three days in H2. Piglets were weighed at birth and ear-tagged individually within the first 24 h post-partum. After this, they were followed until slaughter. Piglets weighing less than 0.9 kg at birth were excluded from the study due to their poor prediction of survival. Piglets weighing 0.9–1.2 kg at birth were categorized as S (*n* = 12 from H1, *n* = 17 from H2) and piglets weighing ≥ 1.3 kg as L (*n* = 18 from H1, *n* = 32 from H2).

### 2.3. Colostrum Sampling

Sow colostrum samples were taken within the first six hours after the first piglet of the litter was born. A piece of cotton with alkaline disinfectant solution (EasyDes/Neo-Amisept) was used to clean two teats and the surrounding skin of the udder. One researcher milked the sample using disposable gloves while another person kept the already born piglets away from the teats for the duration of sampling. The colostrum sample was placed in the refrigerator until immunoglobulin G (IgG) measurement with a Brix refractometer (Atago Master Brix refractometer, 0–53%; Atago, Tokyo, Japan)—a measurement reflecting the colostrum quality in this study. A drop of colostrum was placed on the daylight plate of the refractometer, and the result was read from the scale by viewing through the eyepiece. Reference values for colostrum quality were obtained from Hasan et al. [29]. Colostrum intake of the piglets between birth and the first day of life was calculated for H1 according to a formula by Devillers et al. [30]. In H2, cross-fostering of piglets occurred so early that proper colostrum intake could not be evaluated.

### 2.4. Blood Sampling and Determination of Piglet Serum Immunoglobulins

At the age of one to five days, one blood sample was taken from the jugular vein of each study pig. Samples were stored in the refrigerator overnight and centrifuged at 100× *g* for ten minutes. Serum immunoglobulins were measured the following day after sampling using an immunocrit method described by Vallet et al. [31].

### 2.5. Follow-up of the Study Pigs

Study piglets were checked for cross-fostering by comparing the information collected about the dam at birth to a nursing dam at the moment of blood sampling. If the piglet was suckling another sow at the time of blood sampling, it was marked as being cross-fostered. The pigs were weaned approximately at the age of four weeks into separate weaning departments in the piglet-producing herds. They were transported to finishing units, one for each piglet-producing unit, when they were about nine weeks old. The study pigs were weighed five times during the follow-up period. The average daily weight gain (ADG) of the pigs was calculated for the corresponding production stages. The follow-up period for the study pigs is summarized in Figure 1. Additionally, farm workers were requested to document the medical records for individual study pigs for the entire study period.

### 2.6. Fecal Sampling of the Study Pigs

Fecal samples were taken from the pigs in the weaning unit and again in the finishing unit (Figure 1). The rectal samples were collected with disposable gloves and transferred into individually marked plastic tubes (Sarstedt, Faeces tube 76 × 20 mm, ref 80.734.001, Nümbrecht, Germany), which were placed immediately in a cool box. Samples were moved to −18 °C within one hour and to −80 °C on the following day at the latest.

### 2.7. Study Pig Selection for Fecal Analysis

After all pigs were slaughtered, fecal samples from 30 pigs of 19 sows (8 in H1 and 11 in H2) were selected for lactobacilli analysis. The selection criteria included the following: 1. the dam of the pig had not received any medications during the gestation period before the birth of the study pig, 2. both fecal samples were available from the selected individuals, and 3. both S and L pigs were represented equally in the final sample. One to four piglets per sow fulfilled the selection criteria from H1 and one to two piglets from H2. The allocation procedure resulted in six S and eight L pigs from H1 and 6 S and 10 L pigs from H2. Finally, the selected frozen samples (altogether 12 S and 18 L) were transported to the collaborative laboratory in a cool box within three hours, after which they were stored at −80 °C until lactobacilli analysis.

### 2.8. Determination of Fecal Lactic Acid Bacteria Composition

Fecal samples were thawed on ice overnight. Tenfold dilution series were made using 1 g of feces in 0.85% saline. The serial dilutions 10^−5^, 10^−6^, and 10^−7^ were plated on BL agar (Nissui Pharmaceutical, Tokyo, Japan) and incubated at 37 °C for 72 h, anaerobically (10% H2, 10% CO2, and 80% N2; Concept 400 anaerobic chamber, Ruskinn Technology, Leeds, UK). Colonies were counted and all different looking colonies were picked into Gifu Anaerobic Medium Broth (GAM broth; Nissui Pharmaceutical, Tokyo, Japan) followed by incubation at 37 °C for 24 h. The grown colonies were viewed under a microscope and the non-motile rod-shaped bacteria were streaked on BL agar plates. The plates were incubated anaerobically at 37 °C for 72 h. For DNA isolation, one colony from each plate was suspended into 50 µL TE buffer, heated at 95 °C for 10 min, chilled on ice for 2 min, and centrifuged for 15 min at 17,000× *g*. The supernatant containing the DNA was stored at −20 °C.

To identify different lactobacilli, a colony PCR was performed with lactobacilli strain-specific primers. The PCR was conducted using 1 µL of DNA, 12.5 µL of 2x My Taq HS Red Mix (Bioline, London, UK), 10.5 µL of RNAse free water, and 0.5 µL of 0.2 µM primers Lac1 (5′-AGCAGTAGGGAATCTTCCA-3′) and s-g-Lab-0677-a-A-17 (5′-CACCGCTACACATGGAG-3′) with a final volume of 25 µL. For the PCR conditions, the initial denaturation step was at 95 °C for 1 min; 35 cycles of denaturation at 95 °C for 15 s, primer annealing at 58 °C for 15 s, extension at 72 °C for 15 s, and final extension at 72 °C for 10 min (Applied Biosystems Veriti 96-Well Thermal Cycler, Thermo Fisher Scientific, Waltham, MA, USA). The PCR product was run on a 1% (*w/v*) agarose gel with gel electrophoresis and visualized with ethidium bromide under UV light.

To identify which species of lactobacilli each sample contained, another PCR with 16S primers, 27F (5′-AGAGTTTGGATCMTGGCTCAG-3′), and 1492R (5′-CGGTTACCTTGTTACGACTT-3′) was performed in the PCR conditions of initial denaturation at 95 °C for 1 min; 35 cycles of denaturation at 95 °C for 15 s, primer annealing at 53 °C for 15 s, extension at 72 °C for 15 s, and final extension at 72 °C for 10 min. The PCR product was purified with polyethylene glycol 8000/NaCl (PEG 8000, AppliChem, Darmstadt, Germany). The purified DNA was measured and diluted correctly for sequencing performed by the Institute for Molecular Medicine Finland (FIMM, Helsinki, Finland). The lactobacilli species were identified using the BLAST database of the National Center for Biotechnology Information (NCBI) with 96% accuracy.

### 2.9. Statistical Analysis

SPSS (IBM SPSS Statistics 25) was used in the statistical analyses. The normality of the continuous variables was assessed visually (histogram) and with the Shapiro-Wilk test. Parity (range 1–5) was re-categorized into three categories; 1: gilts, 2: 2nd to 3rd parity, and 3: 4th to 5th parity, respectively. Colostrum quality measures (*n* = 18) did not follow a normal distribution and thus a Mann-Whitney U test was used to test for differences between herds in colostrum quality.

A mixed model was applied to test for differences between herds, piglet size, and piglet serum immunoglobulins. One outlier was excluded from the analysis due to it being clearly beyond the normal distribution of the rest of the data. The model included piglet age (in minutes) as a covariate. Herd and birth size were included as fixed factors in the model. Interaction between herd and birth size was included in the preliminary model but discarded because it was not significant.

Due to skewness of the total lactobacilli count, it was log-transformed for the mixed-model procedure and subsequently back-transformed to present the results. Univariate analyses were used to test single associations between the explanatory variables of interest and total lactobacilli count (Table 1).Variables with a significance level of *p* ≤ 0.25 in the univariate analyses were included in a mixed model with fecal sampling point as a repeated effect. In addition to the variables selected based on the results from the univariate analyses, the fecal sampling point was also included as a fixed factor, and all relevant interactions were tested. The following factors were significantly associated and were thus tested in separate models: average bodyweight (BW) (at the time of fecal sampling) with ADG (during post-weaning and finishing periods); and CI with piglet size at birth (S/L). Non-significant factors (*p* ≥ 0.1) and interactions were excluded from the model one by one. As ADG and BW correlated, a separate model was run with BW, but BW was not significantly associated with the lactobacilli count. Herd, birth size, and their interaction were included in the final model as fixed factors, and ADG was included as a covariate.

Due to limited data, the effects of CI on the total lactobacilli count were tested only for H1 data. In the first step, the sampling point and birth size were included as fixed factors and ADG and CI as covariates. As birth size and CI were correlated, they were tested in separate models. Non-significant factors (*p* ≥ 0.1) were excluded one by one. The final model only included CI.

Model functionality was confirmed by assessing the normality of the model residuals.

## 3. Results

### 3.1. Sow Colostrum Quality, Piglet Immunoglobulins, and Pig Growth

Colostrum quality was numerically better in H1 compared to H2: the median colostrum quality in H1 was 27.5, ranging from 24 to 32, and 25.5 in H2, ranging from 21 to 26. In H1, one sample out of eight (12.5%) was “borderline”, four samples (50.0%) were “adequate”, and three samples (37.5%) were “very good”. In H2, three out of 11 samples (27.3%) were “borderline”, seven (63.6%) were “adequate”, and one sample was omitted due to problems while measuring the Brix value. The mean piglet serum immunoglobulin value was 0.13 (SD 0.03) for both herds. One high piglet serum immunoglobulin value (0.35) was discarded from H1 to normalize the data. As a result of the mixed model, the mean serum immunoglobulin values were numerically larger in H1 and for L piglets, but differences were not statistically significant. Herd, piglet birth size, or piglet age during blood sampling were not associated with the serum immunoglobulin results (*p* > 0.1, for all).

The average weights and ADGs for L and S piglets in both herds are represented in Table 2.

Medical records were obtained from both herds in a reliable manner only prior to fecal sampling point 1. Altogether, five study pigs (one in H1 and four in H2) had received an antimicrobial treatment course before the first fecal sampling point. After that, verifying the information concerning the treatments was not possible.

### 3.2. Fecal Lactobacilli Count of Study Pigs

Total lactobacilli count (CFU/mL) numerically increased in H1 and decreased in H2 over time. Lactobacilli counts in both herds and pig sizes at birth are shown in Table 3.

As a result of the mixed model, the birth size was associated with the total lactobacilli count (F43 = 6.3, *p* = 0.02). Estimated marginal (EM) means for the interaction term herd*size showed that the difference in total lactobacilli count between size categories was seen in H2, where S piglets had a significantly higher lactobacilli count compared to L piglets (F43 = 6.9, *p* = 0.01) (Figure 2). Moreover, S piglets had a higher lactobacilli count in both herds compared to L piglets, although the difference was not significant for H1 (Figure 2). Average daily weight gain tended to be positively associated with the lactobacilli count (*p* = 0.09).

In the model including CI (only H1), CI tended to be significantly associated to the total lactobacilli count (F20 = 4.4, *p* = 0.05).

### 3.3. Diversity of the Fecal Lactobacilli Composition

Altogether seven lactobacilli were identified in the feces of the study piglets in decreasing order of prevalence: *Limosilactobacillus reuteri*, *Lactobacillus johnsonii*, *Limosilactobacillus pontis*, *Lactobacillus amylovorus*, *Lentilactobacillus parabuchneri*, *Lactiplantibacillus plantarum*, and *Limosilactobacillus vaginalis*. The number of lactobacilli spp. in individual pigs at fecal sampling point 1 ranged from one to five (median 2) in H1 and from zero to two (median 1) in H2, respectively. At fecal sampling point 2, the numbers of lactobacilli spp. ranged from zero to three (median one) in H1 and from zero to one (median one) in H2, respectively.

In H1, the number of lactobacilli spp. was significantly higher than in H2 at both fecal sampling points (Mann-Whitney U test, *p* = 0.01). All seven lactobacilli spp. were present in H1, whereas only two species were present in H2 (Figure 3 and Figure 4). Of all lactobacilli recognized, 24% of lactobacilli spp. remained unidentified at fecal sampling point 1 and 72% at fecal sampling point 2. Overall, 57% of all lactobacilli spp. were not successfully sequenced with qPCR and therefore remained unidentified.

*L. reuteri* was numerically the predominant lactobacilli at both fecal sampling points and in both herds. In H1, the abundance of *L. reuteri* increased between both fecal sampling points and in both size categories. Inversely in H2, the abundance of *L. reuteri* decreased between both sampling points and in both size categories. *L. johnsonii* was another lactobacilli sp. identified at both sampling points and in both size categories in H1, but with decreasing abundance. In H2, *L. johnsonii* was identified only in L piglets at fecal sampling point 1.

All other lactobacilli were only identified in piglets from H1. *L. pontis* was identified in S piglets only at fecal sampling point 1 and in L piglets at both sampling points with a decreasing abundance. The abundance of *L. amylovorus* increased in S piglets between sampling points and was only identified in L piglets at fecal sampling point 2 and altogether at a lower abundance compared with S piglets. *L. parabuchneri*, *L. plantarum*, and *L. vaginalis* were identified occasionally and only at fecal sampling point 1. Moreover, *L. parabuchneri* and *L. plantarum* were identified only in S piglets, whereas *L. vaginalis* was identified only in L piglets.

## 4. Discussion

Herd-level factors contributed to the lactobacilli count and species diversity of growing pigs, with the latter being more prominently influenced. Previous research supports the fact that *Lactobacillus* is a predominant genus in pig GITs and an abundance of *Lactobacillus* species plays a pivotal role in modulating the health status of the host [26]. Two predominant species, *L. reuteri* and *L. johnsonii*, were identified in both herds. However, differences in species diversities were found despite gut microbiota stabilizing around the time of weaning. Piglet size at birth seems to be one factor influencing the development of GIT lactobacilli composition. In addition, colostrum intake was associated with the GIT lactobacilli count of growing pigs, suggesting a quite powerful and long-term influence of colostrum.

### 4.1. Colostrum and Serum Immunoglobulins

In this study, we found that colostrum quality was numerically better in H1 compared to H2. Even though the median values for colostrum quality were at “adequate” levels in both herds, as defined according to Hasan et al. [29], “very good” values were only obtained in H1. Unsurprisingly, piglet serum immunoglobulins were numerically higher in H1 compared to H2. Moreover, H1 pigs had more diverse fecal lactobacilli content than H2 pigs. Bian et al. [7] found the lactose concentration of sow milk to correlate with lactobacilli abundance in their piglets. We did not determine colostrum composition in detail in our current study. Overall, sows are not able to nurse all their progeny and cross-fostering of piglets has therefore become a management routine in farrowing units. All piglets do not receive colostrum from their own dams. In our data, cross-fostering did not correlate with the total lactobacilli count. Further investigation is needed, as other factors linked to colostrum could also have contributed to the total lactobacilli count in our pigs, not only colostrum immunoglobulin content. Our results did not show any significant difference in piglet serum immunoglobulins between S and L piglets at birth or over time. Conversely, Schnier et al. [32] reported numerically lower immunocrit values in S piglets compared to L piglets. On the other hand, Morton et al. [33] reported that piglet birth weight did not affect serum immunocrit values, which is in line with our results.

### 4.2. Growth of Study Pigs and Lactobacilli Count in Fecal Samples

To the knowledge of the authors, more attention has so far been paid to lactobacilli diversity compared to lactobacilli count. Our results showed an increase in total lactobacilli count in H1 and a decrease in H2 between sampling points. Lactobacilli was significantly higher at the second fecal sampling point in H1. Based on our observations, H1 was better organized in terms of management routines and the hygiene level of piglets and sows. For example, piglets were mixed notably less frequently.

Despite our study being carried out during early production phases (until 35 days of age), our results are in line with a previous finding where S pigs remain small in the long term and their average daily gains are lower compared to pigs born with normal bodyweights [16]. Interestingly, the overall growth of the H1 study pigs was greater than in H2 study pigs during the pre- and post-weaning periods. However, when transitioning to the finishing phase, the growth of the H2 study pigs increased in S and L pigs, resulting in higher ADGs for both size categories compared to L pigs in H1. Additionally, the lactobacilli count in H2 peaked significantly between the first and second fecal sampling points. At the finishing phase, we observed notably better hygiene practices in H2 compared to H1, which may have contributed to this result.

In a study by Li et al. [16], piglets born with normal weight (>1.1 kg) had higher *Lactobacillus* counts between 7 and 35 days of age than piglets with lighter birth weights (<1.1 kg), which in turn had a different gut microbial community and metabolic status compared to their normal-birth-weight siblings. The researchers [16] therefore speculated that these factors may contribute to the impaired growth and development of S piglets. Results from a study by Gaukroger et al. [28] also support the hypothesis that GIT *Lactobacillus* composition is associated with the growth performance of piglets. ADG tended to be associated with the total lactobacilli count in our study pigs, but we were not able to model factors associated with the lactobacilli spp. diversity because of uncertainties regarding sequencing. The sequencing technique is one source of bias in our present study, as quite many lactobacilli remained unidentified.

### 4.3. Lactobacilli Diversity in Samples

More lactobacilli spp. were identified in H1 at both fecal sampling points and similar lactobacilli spp. were the most abundant in both herds: *L. reuteri* and *L. johnsonii*. Lactobacilli are present throughout the GIT, but principally exhibit higher levels in fecal samples [26,34]. Further, Holman et al. [34] observed that the sampling point within the GIT causes the largest variation in the microbiota and that the core microbiota of swine include *Lactobacillus* species. Moreover, *Lactobacillus* has been reported to be the most dominant genera in pig GITs from birth until 56 days of age [28]. Piglet size may influence the GIT lactobacilli content even though the effect of birth weight remains controversial. Our results indicated a difference in fecal lactobacilli composition between S and L piglets. Li et al. [29] found S piglets to have greater *Lactobacillus* spp. diversity in their ileal contents at the age of 35 days (35.7%) and in their colons between the age of 7 and 35 days (62.9%). According to Li et al. [16], birth weight was not significantly associated with fecal bacterial richness and diversity at any tested time point from birth until 35 days of age. However, the Shannon diversity index and the number of operational taxonomic units (OTUs) seemed to be higher in S piglets compared to piglets with normal birth weights between 3 and 35 days post-partum [16]. Recently, Gaukroger et al. [28] showed that piglet birth weight had a significant effect on microbiota richness between 21 to 56 days of age. In a human study by Jia et al. [35], higher birth weight was associated with an increased level of *Lactobacillus* in the pre-term group and increased bacterial diversity in the full-term infant group. Pre-term infants had significantly more lactobacilli in their meconium compared to full-term infants [35]. Li et al. [16] have reported that piglet age was a significant factor affecting the overall gut bacterial composition irrespective of birth weight. In human infants, microbiota variability occurs over time and is different in pre-term compared to full-term infants [35].

Kim et al. [36] showed that bacterial community difference was most significant between growers and finishers corresponding to the production phases of the present study. Gaukroger et al. [28] reported that piglet age was a significant key driver in the development of microbial community composition—the most stable and uniform microbiota was observed from 35 days of age onwards compared to microbiota development during the first week of life. This is in line with a study by Unno et al. [37], in which microbiota stability was reported to be low at an early production stage (zero to seven days) followed by stabilization at four weeks of age. Based on these findings, we can assume that the microbiota of our study pigs had already stabilized at the time of both fecal samplings. During the stabilization process, the GIT becomes colonized with exogenous bacteria and goes through significant ecological changes both in structure and diversity [37]. Many factors including genetics [3,7,38], diseases, antimicrobials, feed, and management (hygiene, housing conditions, and cross-fostering habits) are additionally known to influence GIT microbiota [3,7,38].

*L. reuteri* was the predominant lactobacilli sp. identified in this study in both herds and over time. *L. reuteri* is known to be acid-tolerant and resistant to swine bile and to treatment with chlortetracycline in vitro [18]. Treatment with *L. reuteri* resulted in a richer and unique jejunal microbiome when compared between control and antimicrobial treated pigs, and no significant effect of treatment was observed on the cecal and colonic microbiome [18]. Moreover, lactic acid bacteria were found to help in the recovery of diarrheal piglets, which were experimentally infected with enterotoxigenic Escherichia coli (ETEC) [39]. Yang et al. demonstrated that strain-specific *L. reuteri* administered to piglets colonizes the intestinal mucosa and improves the cecal microbiota profile and whole-body antioxidant and immune status, leading to better growth and lower morbidity and mortality rates [40]. The following five *Lactobacillus* spp. with higher percentages were found in recovered piglets compared to diarrheal ones: *L. reuteri*, *L. amylovorus*, *L. acidophilus*, *L. johnsonii*, and *L. crispatus* [39]. Supported by these studies, *L. reuteri* currently has promising potential as a probiotic candidate and strain-level investigations are warranted. Four out of five lactobacilli highlighted by Bin et al. [39] were also found in our study pigs. Therefore, their potential as a probiotic product should not be ignored, and further studies are needed.

Certain factors were out of the scope of this study, such as detailed investigation of feed composition and antimicrobial treatments of the study pigs, which could potentially have influenced the results. Further, weaning practices on farms and transportation to the finishing units may have influenced the results as well. Konstantinov et al. [1] found weaning to be accompanied by a significant reduction in lactobacilli. However, we investigated the fecal microbiota of the pigs right before they were transported to the finishing units, which usually occurs more than one month after weaning, or to the slaughterhouse, which similarly happens at least three months after transportation to the finishing unit. Consequently, a study including fecal microbiota investigation throughout the production chain is needed to overcome these unexplained issues and to reliably assess the dynamics of the microbiota between different production phases.

## 5. Conclusions

Both herd-level and individual differences in the fecal lactobacilli dynamics of growing pigs were identified in the present study. Regardless of the perceived poorer management practices in herd 1 as compared to herd 2, lactobacilli diversity was significantly higher in this herd at both sampling points. Small piglets had a numerically higher lactobacilli count throughout the study and this difference was significant in H2. In summary, herd-level factors clearly contribute to the microbiota of pigs but birth weight also plays a potential role in the GIT lactobacilli dynamics. Our present study strengthens the assumption of the importance of colostrum as a contributor to fecal lactobacilli composition of the progeny, and this influence seems to be long term. Therefore, the associations between maternal immunity, pig microbiota, and performance are encouraged to be investigated more thoroughly.

## Figures and Tables

**Figure 1 animals-11-00113-f001:**
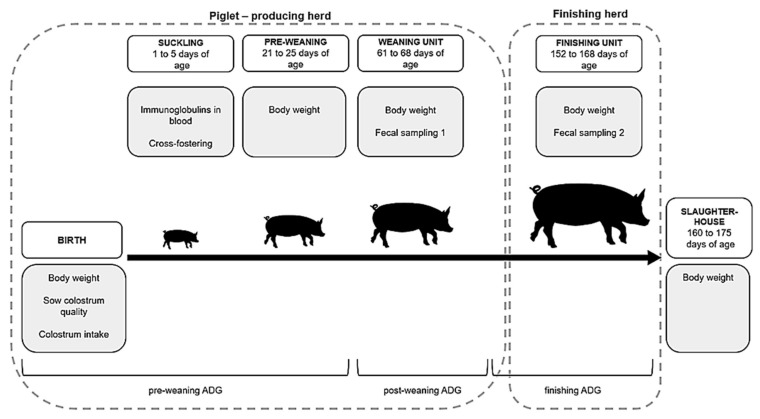
Follow-up of the study pigs from birth to slaughter in two herds (H1 and H2). White boxes represent production phases and age variation of study pigs at corresponding points. Gray boxes illustrate the procedures performed at each phase. Study pigs were transported from the piglet-producing herds to the finishing herds between fecal sampling points 1 and 2. ADG = average daily weight gain (g).

**Figure 2 animals-11-00113-f002:**
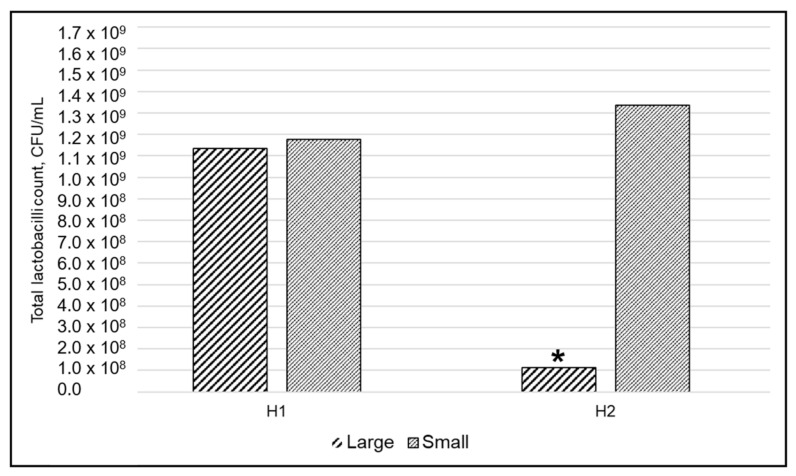
Back-transformed EM means of the lactobacilli count in piglets born small or large in two herds (H1 and H2) analyzed by mixed model (*n* = 30). ***** Significant difference between small and large piglets was found in H2 (*p* = 0.01).

**Figure 3 animals-11-00113-f003:**
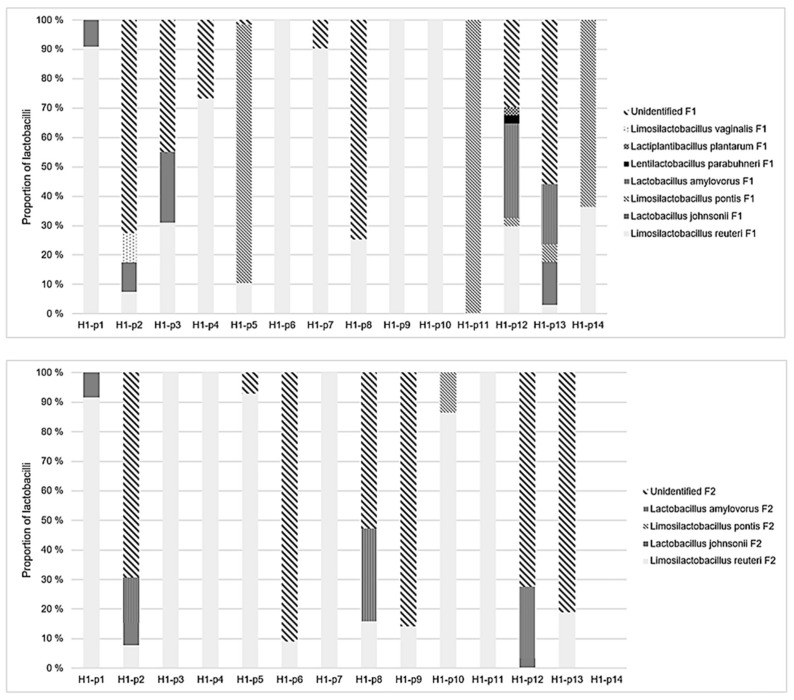
The proportion of lactobacilli species identified from the feces of individual pigs (14 pigs, p1–14) in the weaning unit (upper picture, fecal sampling point 1 (F1)) and fattening unit (lower picture, fecal sampling point 2 (F2)) in herd 1 (H1).

**Figure 4 animals-11-00113-f004:**
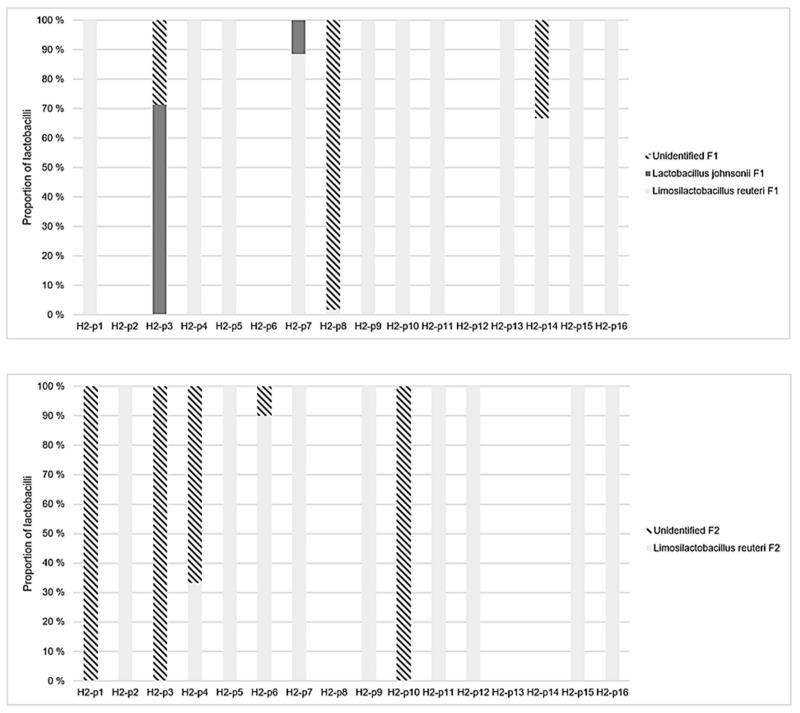
The proportion of lactobacilli species identified from the feces of individual pigs (16 pigs, p1–16) in the weaning unit (upper picture, fecal sampling point 1 (F1)) and the fattening unit (lower picture, fecal sampling point 2 (F2)) in herd 2 (H2). The number of lactobacilli spp. was significantly lower in H2 compared to H1 at both fecal sampling points (Mann-Whitney U test, *p* = 0.01).

**Table 1 animals-11-00113-t001:** Testing of factors associated with total lactobacilli count (CFU/mL) in the fecal samples of growing pigs, univariate analysis. * Variables with a *p*-value < 0.25 were tested in the multivariable analysis.

Variables, Univariate	Test	*p*-Value
Sow parity, categorical	ANOVA	0.27
Colostrum quality, %	Spearman correlation	0.59
* Piglet size at birth, small/large	*t*-test	0.04 *
* Colostrum intake, g	Pearson correlation	0.19 *
Cross-fostering, yes/no	*t*-test	0.28
Piglet serum immunoglobulins, ratio	Spearman correlation	0.48
* Bodyweight of a pig at fecal sampling points, kg	Pearson correlation	0.08 *
* Average daily gain between fecal sampling points, g	Pearson correlation	0.09 *

**Table 2 animals-11-00113-t002:** Average bodyweight (BW), average daily weight gain (ADG), and their standard deviations (SD) of piglets born either small (S) or large (L) (*n* = 30) at different time points.

Parameter	Herd 1			Herd 2		
	Small	Large	Total	Small	Large	Total
*n*	6	8	14	6	10	16
	Average (SD)	Average (SD)	Average (SD)	Average (SD)	Average (SD)	Average (SD)
Birth BW, kg	1.1 (0.0)	1.6 (0.2)		1.1 (0.1)	1.6 (0.1)	
Pre-weaning ADG, g	203.2 (32.5)	208.0 (42.4)	206.0 (9.9)	161.1 (35.7)	208.2 (45.2)	190.8 (11.7)
Weaning BW, kg	6.0 (1.0)	6.5 (0.9)		4.7 (1.0)	6.3 (1.1)	
Post-weaning ADG, g	322.2 (110.7)	376.8 (80.6)	353.4 (25.4)	280.1 (106.7)	330.2 (91.4)	311.4 (24.3)
Post-weaning BW, kg	19.9 (5.7)	22.7 (3.5)		16.1 (4.9)	20.4 (4.5)	
Finishing ADG, g	821.4 (41.6)	907.3 (113.1)	870.5 (26.1)	987.1 (97.5)	1044.1 (132.5)	1022.8 (30.1)
Finishing BW, kg	91.3 (7.4)	101.6 (12.4)		119.0 (13.8)	127.3 (15.3)	
Slaughter BW, kg	114.4 (11.9)	113.0 (8.5)		117.1 (16.9)	123.0 (13.0)	

**Table 3 animals-11-00113-t003:** Descriptive statistics of total lactobacilli count in 30 study pigs divided by fecal sampling point, herd, and bodyweight category based on their birth weight. H1 = herd 1, H2 = herd 2, S = pigs born small (0.9–1.2 kg), L = pigs born large (≥1.3 kg). Fecal sampling point 1 occurred when pigs were in the weaning unit and fecal sampling point 2 when pigs were in the finishing unit.

Fecal Sampling Point 1					
Herd, body weight category	Median, CFU/mL	Min, CFU/mL	Max, CFU/mL	25th percentile, CFU/mL	75th percentile, CFU/mL
H1, S	1.74 × 10^9^	1.00 × 10^8^	7.50 × 10^9^	3.10 × 10^8^	3.40 × 10^9^
H2, S	3.45 × 10^9^	0.00	2.18 × 10^10^	2.60 × 10^8^	6.00 × 10^9^
H1, L	1.30 × 10^9^	1.00 × 10^8^	1.34 × 10^10^	2.60 × 10^8^	3.45 × 10^9^
H2, L	1.35 × 10^8^	0.00	1.40 × 10^9^	1.00 × 10^7^	8.00 × 10^8^
**Fecal Sampling Point 2**					
H1, S	1.40 × 10^9^	2.40 × 10^8^	3.80 × 10^10^	2.90 × 10^8^	3.80 × 10^9^
H2, S	4.50 × 10^8^	0.00	1.60 × 10^9^	2.00 × 10^7^	1.20 × 10^9^
H1, L	7.65 × 10^8^	0.00	1.10 × 10^11^	1.00 × 10^8^	5.65 × 10^9^
H2, L	1.50 × 10^7^	0.00	1.00 × 10^9^	6.00 × 10^6^	3.00 × 10^8^

## Data Availability

All data used in the current study are available from the corresponding author on reasonable request.

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
