# Peer review of "Herd-Level and Individual Differences in Fecal Lactobacilli Dynamics of Growing Pigs"

_animals, 2021, doi:10.3390/ani11010113_

Round 1

Reviewer 1 Report

The manuscript presented for the review undertakes an important and still poorly understood issue of farm animals gastrointestinal tract microbiome.

I have some remarks which are as follows:

In Materials and Methods part the authors state that the animals were transported from the weaning unit to finishing unit - what was the distance between these units? Were these two separate farms? It is important as the environment may also affect intestinal microbione, so it would be difficult to compare samplings 1 and 2 in case of both herds.

In line 109 the authors mention about "hygiene standards" - how were they evaluated?

Lines 116-117 - I also see that the feeding differed in some period, this also is a factor affecting microbiome.

Were the colostrum samples examined for immunoglobulins only? The authors use the term "colostrum quality" which may suggest something more.

Why was colostrum intake recorded only from herd 1?

Table 3 - the authors present sow parity as a variable - it was not mentioned in Materials and Methods part.

Reviewer 2 Report

L2: title does not reflect the work properly. Please change to an informative title.

L17-19: rewrite please. This can be said simply like this ‘ Selection for hyper-prolific sows has led to increased litter size, decreased birth weight, and increased within-litter variation.’

L25: rewrite please. Sentence not completed.

L34: please mention the replicate per group? n=?

L38-39: the quick mentioning of the difference between H1 and H2 is needed here.

L44: if P=0.05, then the association is not significant. It’s a trend.

L47-67: it is not easy to make a connection between finding antibiotic alternative with the aim of this study which is looking at growth variation. What connection is there between replacing antibiotic and growth variation?! It becomes even more difficult to link those two while in this study when authors mention in MM that “…No oral antimicrobial agents or growth promoters were given to any of the study pigs in the feed…”. After reading the abstract, the expectation is that the introduction is going to start with growth variation and how it might be linked to Lactobacillus count and diversity. It might be good idea to reshuffle the information a bit e.g. bringing the forth and third paragraphs come before the first two which of course it’s going to change the order others as well.

L110-111: please review” … herd of H1 appeared to have lower hygiene…”?? H1 cannot be better managed and low in hygiene at the same time!

L120-125: did you select one piglet from each sow? If yes, the difference in the piglet weight could be simply due to the differences between sows.

L125: n=12 and 18 from each herd or both herd together?

L121: “ … were born to ..” change to from

Table 1: please run statistics on the performance data to see if the performance between S and L are significantly different. I guess, sow’s effect should be considered as a random effect in the model as piglets are from different sows.

Table 2: please report the mean and SE for the groups. Also, statistical comparisons should be done. If there is no significant difference between H1 and 2, it may be a good idea to pool the data from these to herd to get better statistical power.

Figure 2. please mention P value in the figure if any

Figure 4: the same as above. Figures should be self-explanatory. Readers should not go to the text to find P values. They need to be provided on the figures.

Conclusion: this section is a stand alone section when you mention H1 or H2, they mean nothing. Please mention the difference between them so that the reader can see the link between the change in bacteria and herd management.
